# Tactile Image-to-Image Disentanglement of Contact Geometry from Motion-Induced Shear

**Anupam K. Gupta**
Department of Engineering Mathematics
Bristol Robotics Laboratory
University of Bristol, U.K.
`anupam.gupta@bristol.ac.uk`

**Laurence Aitchison**
Department of Computer Science
University of Bristol, U.K.
`laurence.aitchison@bristol.ac.uk`

**Nathan F. Lepora**
Department of Engineering Mathematics
Bristol Robotics Laboratory
University of Bristol, U.K.
`n.lepora@bristol.ac.uk`

**Abstract:**

Robotic touch, particularly when using soft optical tactile sensors, suffers from distortion caused by motion-dependent shear. The manner in which the sensor contacts a stimulus is entangled with the tactile information about the stimulus geometry. In this work, we propose a supervised convolutional deep neural network model that learns to disentangle, in the latent space, the components of sensor deformations caused by contact geometry from those due to sliding-induced shear. The approach is validated by showing a close match between the unsheared images reconstructed from sheared images and their vertical tap (non-sheared) counterparts. In addition, the unsheared tactile images faithfully reconstruct the contact geometry masked in sheared data, and allow robust estimation of the contact pose of use for sliding exploration of various planar shapes. Overall, the contact geometry reconstruction in conjunction with sliding exploration were used for faithful full object reconstruction of various planar shapes. The methods have broad applicability to deep learning models for robots with a shear-sensitive sense of touch.

**Keywords:** Robotic Touch, Disentanglement, Shear, Object Reconstruction

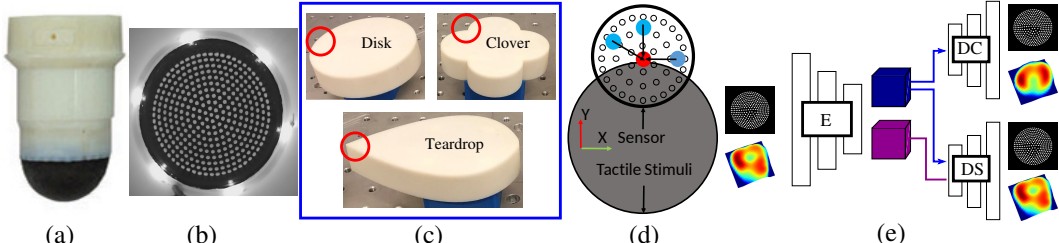

Figure 1: **Experiment setup and model architecture.** (a) TacTip sensor, (b) Sensor internal surface, (c) 3D-printed tactile stimuli with approximate locations of data collection (red circles), (d) Schematic of data collection, showing target (red dot) and initial contact locations (blue dots). Canonical (tapping) data was collected by vertically contacting the target; sheared (sliding) data was collected by sliding to the target, and (e) Schematic of the *Disentangled UnshearNet* model architecture with representative input/output tactile images and contact geometry reconstructions.

## 1 Introduction

A fundamental aspect of tactile sensing is that it measures the environment via physical contact. This entangles the stimulus related tactile information with that induced by the manner of contact between the sensor and the stimulus. For example, when sliding or rubbing fingers across tactile stimuli,

5th Conference on Robot Learning (CoRL 2021), London, UK.

soft tactile sensors are inevitably distorted by motion-dependent shear, making the sensor response history-dependent and generalization hard in tactile tasks. This issue is prevalent in camera-based optical tactile sensors that use the lateral motion of markers to infer contact geometry, such as the TacTip [1, 2], ChromaTouch [3], GelForce [4] and the optical tactile sensor described in [5], but may occur also in other shear-sensitive tactile sensors. Generally, this shear distortion of the sensor response when sliding along surfaces would make it difficult to perform tactile-dependent tasks such as shape reconstruction and continuous object exploration or manipulation.

Therefore, an important capability is to separate the deformation due to the manner of contact from that due to the stimulus geometry. For example, when continuously exploring an object surface, the sliding-induced shear would need to be removed from tactile image to extract the local contact shape. Previous work on extracting pose information from contact to continuously slide a tactile sensor over complex objects used a deep convolutional neural network trained to map between shear-distorted inputs and the desired (pose) outputs, which was necessary for sliding exploration over a range of complex novel objects [6, 7, 8]. Here, we suggest, an alternate disentanglement based approach that learns to remove motion-induced shear from raw sheared sensor response that has several advantages over previous task-specific approach of training insensitivity to shear, in that: (a) it is computational and data efficient by only requiring the 'unshearing' of the tactile data to be learnt once instead of relearning for each tactile task, such as when applied to full object reconstruction (Sec. 4.4); (b) the models can then be learnt more easily on unsheared data, which we demonstrate by predicting local object pose; moreover, in some cases no model need be learnt at all, which we demonstrate with contact surface reconstruction using a Voronoi tessellation from the tactile marker data; and (c) the methods are extensible to including other downstream variables of interest appropriate to the task, rather than having to completely retrain new models from scratch.

Our approach to disentangling environmental factors of variation and object attributes in tactile sensing is based on work in computer vision where disentanglement is considered important to improve the generalizability of machine learning models, for example to aid in learning of robust object representations for object classification under novel scenarios [9]. Inspired from this, here we propose a supervised convolutional neural network model that learns to disentangle the components of sensor deformation caused by the contacted object geometry from those due to motion-induced shear. The models were trained using paired data from vertical tap (non-sheared) and continuous (sliding) touch, with an image-to-image *Disentangled UnshearNet* model architecture that outputs unsheared tactile images from sheared images (Fig. 1). Our validation: (1) quantifies that the unsheared images from continuous motion match their paired tactile images from discrete taps (Sec. 4.1); (2) demonstrates that local contact geometry reconstructed from unsheared images using Voronoi match their paired tapping counterparts over several stimuli shapes (Fig. 4, top row); (3) infers pose from the sheared tactile images by re-using a convolutional neural network model trained on vertical tap (non-sheared) data, which was then applied to controlling robust sliding exploration around several planar objects (Fig. 4, second row); and (4) combines the two previous steps to attain faithful full-object reconstruction of several planar shapes (Fig. 4, third row).

## 2 Background

Measuring strain is a fundamental component of tactile sensing, as emphasised by Platkiewicz, *et al* [10]: 'the pressure distribution at the surface of a tactile sensor cannot be acquired directly and must be inferred from the [strain] field induced by the touched object in the sensor medium.'

There are many mechanisms whereby normal strain and shear strain are measured by tactile sensors. Taxel-based tactile sensors such as the iCub fingertip [11] and BioTac [12] directly measure normal strain. Nevertheless, this measurement of normal strain will be affected by shearing of the soft sensor medium, although the effects on sensor output will be much smaller than those from normal pressure.

Camera-based optical tactile sensors can be very sensitive to shear because they transduce deformation of the soft sensor surface into a lateral image. For optical tactile sensors such as the TacTip biomimetic optical sensor used here [1, 2, 14] and other marker-based sensors [3, 5], the surface shear will also distort the apparent geometry of the contact. This distortion is because the normal indentation is

represented in the local shear strain of the markers, upon which the global shear strain is superimposed. This contrasts with optical tactile sensors that reconstruct the indentation field from reflection, such as the GelSight [15], where shear will shift the imprint on the tactile image but otherwise the imprint shape should be relatively unaffected.

This effect of shear complicates tactile perception under general motions against an object, and restricts the application of touch on challenging tasks such as tactile exploration and surface shape reconstruction. Studies have been limited to discrete contacts where the sensor taps discretely over the object to minimize shear [17, 18, 20], or to rigid planar tactile sensors [21, 22] that do not shear.

A recent body of work on the TacTip sensor used here focused on mitigating the effects of motion-dependent shear during tactile servo control to slide around planar shapes [6, 19] and complex 3D objects [7, 8]. The shear-dependence problem was addressed in three studies that used a deep convolutional neural network to predict pose by training insensitivity to shear into the prediction network [6, 7, 8]. While effective, this has a drawback that procedures for learning shear-invariance needs to be introduced into the training for each individual task, and thus would not generalize easily to new tasks. Another study did not use deep learning, but instead dimensionally reduced the tactile data [19], where the the pose components naturally separated from the shear components.

This work pursues an alternate approach using deep learning to remove the effect of motion-dependent shear from the sheared tactile image, offering several advantages that were summarized in the introduction. Our approach to disentangling environmental factors of variation and object attributes in tactile sensing is based on work in computer vision where disentanglement is considered important to improve the generalization of machine learning models, for example to aid in learning of robust object representations for object classification under novel scenarios [9]. Once disentangled, different environmental factors and object attributes can be recombined to generate coherent novel concepts thus extending knowledge to previously unobserved scenarios [9]. For example, synthesizing sensor responses to novel stimuli by combining the stimuli attributes and factor of variation without actually observing them to plan action under previously unobserved conditions. In the case of tactile sensing considered here, the stimuli attributes could be novel contact geometry and the factors of variation the motion-induced shear.

## 3 Methods

### 3.1 Experimental Setup

The setup used in this study was similar to that previously used in [6, 7, 8, 19] where a biomimetic soft optical tactile sensor – the TacTip – was mounted on an industrial robot arm (ABB IRB120) as an end effector. The sensor skin morphology mimics the layered dermal and epidermal structure of the human fingertip [1, 2, 14] and encodes tactile information in the shear displacement of the papillae (marker) pins, on the inner side of the sensing surface, caused by the skin deformation. The TacTip used in this work consisted of a 3D printed soft rubber-like hemispherical dome (40 mm radius, TangoBlack+) with its inner surface covered with 331 marker pins with white tips arranged in a concentric circular grid. The dome was filled with an optically-clear silicone gel to achieve compliance akin to human fingertip. An RGB camera (ELP 1080p module) captures the inner surface of the dome. For more details, we refer to the references above. The manual final assembly of the sensor introduces minor variability in the sensor response of the otherwise similar sensors.

For training, we used three 3D-printed (ABS plastic) planar shapes: a circular disk, clover and teardrop shown in Fig. 1 (c) (red circles show the approximate location of data collection). For testing, we used sever planar shapes with various morphologies including two acrylic shapes with distinct frictional properties to the other five 3D-printed shapes (Fig. 4, second row). All shapes were securely fastened to the workspace to prevent accidental motion.

### 3.2 Data Collection

#### 3.2.1 Paired Canonical and Sheared Data

This work proposes a supervised deep learning model *Disentangled UnshearNet* to remove the effect of motion-induced global shear in the tactile images, caused by the friction between the sensor skin

and its contacted surface during contact. We emphasise this motion-induced shear is different from the skin deformation due to local shear from the geometry of the stimulus imprint.

Our method requires paired tactile data with minimal global shear taken from vertically tapping onto the stimulus (referred to as canonical data) and sliding data with random global shear (referred to as sheared data). The sheared data is collected after sliding randomly and is paired with a tactile image from the canonical data by using the same relative poses between the sensor and the stimuli (Fig. 1 (d)). The raw tactile data was cropped, sub-sampled to a $256 \times 256$-pixel region and finally converted to binary images using an adaptive threshold to minimise sensitivity to internal lighting conditions.

In total, data for 200 distinct poses were collected for each of the three training stimuli (Fig. 1 (c)). To generate poses, the sensor location relative to the target contact location (red circle, Fig. 1 (d)) was sampled randomly from a uniform distribution spanning the range [-5, 5] mm in the two lateral directions (along the $x$- and $y$-axes), [-45, -45] deg in yaw ($\theta n_Z$) and [-6, -1] mm in depth ($z$-axis). Ranges of the target poses and perturbations to generate canonical and sheared data were chosen to ensure safe contact of the sensor with the stimulus (Fig. 1 (c)).

**Canonical Data:** The canonical data was taken by vertically tapping onto the stimulus, to give a representative sample of tactile data with minimal global shear just due to the stimulus geometry. This gives target (reference) data that the sheared data from sliding contacts should be restored to. The dataset, in total, had 30,000 samples: 50 instances of each of the 200 poses recorded previously for each of the three stimuli shapes (Fig. 1 (c)). To generate instances, the indentation depth ($z$-axes) was varied randomly between [-1, 1] mm from the original indentation depth.

**Sheared Data:** The sheared data accords to a canonical tactile image distorted by global shear from lateral sliding motion of the sensor on the stimulus. Thus, the sheared data does not give contact-only information akin to canonical data that represented the geometry of the contacted stimulus, but has a significant component due to the contact motion.

The dataset, in total, had 90,000 samples: 150 instances of each of the 200 random poses recorded previously for each of the three stimulus shapes (Fig. 1 (c)). To generate instances for each pose, the lateral position offset for sliding to the target pose was varied randomly between [-2.5, 2.5] mm in lateral direction(s) (along the $x$- or $y$-axes or both), using the procedure shown in Fig. 1 (d).

**Training and Test Data:** The canonical and sheared datasets were partitioned randomly into training, validation and test sets (ratio 60:20:20); i.e. from the 200 unique poses for each stimulus shape (Fig. 1 (c)), 120 poses were assigned to a training set and 40 poses each to validation and testing sets. Overall the canonical and sheared training set contained 18,000 and 54,000 samples respectively, while the validation and testing sets contained 6,000 and 18,000 samples respectively. To generate paired canonical and transformed data for training, each transformed tactile image was paired randomly with one of the canonical instances having the same pose.

### 3.2.2 Pose-labelled Data

To train pose prediction network *PoseNet*, a separate dataset was collected instead of reusing the dataset used to train *Entangled & Disentagled UnshearNet*. This was done to prevent the model from overfitting the training set and ensure performance of the trained *UnshearNet* models despite potential dataset shifts due to a host of factors, including drift in sensor characteristics over time or with continued use due to wear and tear of the tactile sensing surface.

The dataset had, in total, 30,000 samples: 10,000 samples from each of the three training object shapes (Fig. 1 (c)). To generate contact poses, the sensor location relative to the initial contact location was sampled randomly from a uniform distribution spanning the range [-5, 5] mm laterally ($x$- and $y$-axis directions), [-45, -45] deg in yaw ($\theta n_Z$) and [-6, -1] mm in depth ($z$-axis). Ranges of the target poses, as before, were chosen to ensure safe contact of the sensor without causing damage. As discussed above, all tactile images were cropped, sub-sampled and thresholded to $256 \times 256$-pixel to binary (black/white) tactile images. The dataset was partitioned randomly into training , validation and test sets in the ratio 60:20:20 as above, containing 18,000, 6,000 and 6,000 samples respectively.

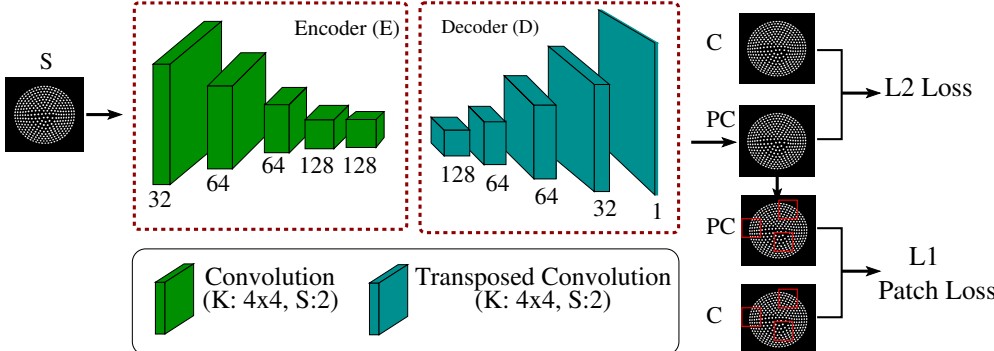

Figure 2: **Entangled UnshearNet architecture:** A shear-distorted tactile image *S* was processed by the encoder *E* and decoder *D* with output *PC*: the unsheared counterpart of input *S* post removal of sliding-induced distortion. The match between predicted unsheared data *PC* and canonical data *C* was enforced by training with a direct comparison between *PC* and *C* along with comparing corresponding patches ($20 \times 20$ pix.) taken from *PC* and *C* to enforce local compliance.

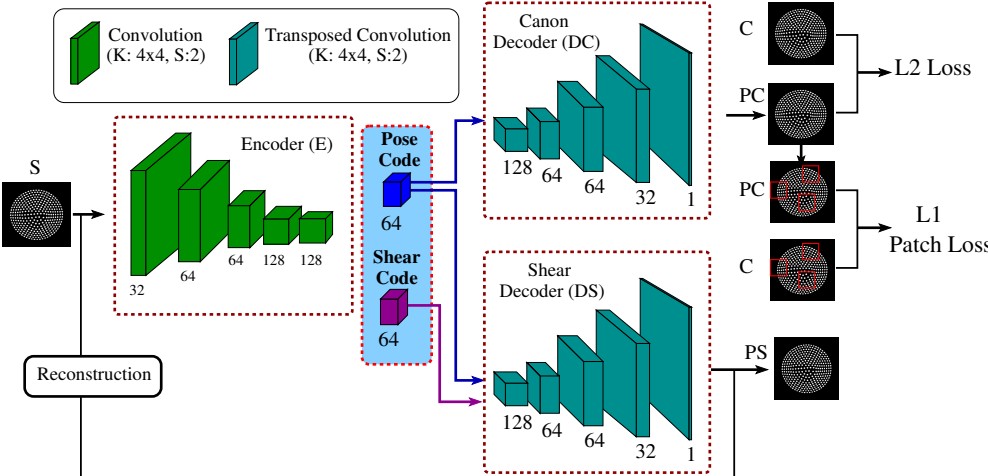

Figure 3: **Disentangled UnshearNet architecture:** Labelling as in Fig. 2, with two decoders *DC* and *DS*. The first decoder has output *PC*: the unsheared counterpart of input *S* post removal of sliding-induced distortion; the second decoder *DS* reconstructed the input *S*, denoted *PS*.

### 3.3  Image-to-Image Models for Shear Removal

To achieve our goal of removing the effects of shear induced by the sliding motion, we trained two models based on the encoder-decoder framework – *Entangled UnshearNet* (Fig. 2) and *Disentangled UnshearNet* (Fig. 3). The former model (baseline) learns the mapping between the input sheared data and its corresponding canonical (tap) data directly. The later model instead, first disentangles the contact-only (pose) sensor response component from the sliding-induced global shear component in the latent space of the encoder *E*; followed by accurate reconstruction of the input sheared image and its corresponding canonical (tap) image using decoders *DS* (sheared or sliding) and *DC* (canonical or tapping). We hypothesized that a more robust mapping between sheared and canonical (tap) data can be learned by disentangling the contact-only and sliding-induced components of sensor response. The mathematical formulation of the *Disentangled UnshearNet* is listed below:

The encoder *E* splits the latent space into two components $\mathbf{z} = (\mathbf{z_p}, \mathbf{z_s}) \in Z_p \times Z_s$, given paired canonical $\{x_c\}_{i=1}^N$ and shear-transformed training samples $\{x_s\}_{i=1}^N$, where $x_c \in X_c$ and $x_s \in X_s$ are in the sets of canonical and sheared data, respectively.

The decoder *DC* takes as input only the pose part of the latent code $\mathbf{z}_p$ and reconstructs the canonical sample corresponding to the input sheared sample. In combination with the encoder *E*, it thus learns a mapping $M : X_t \rightarrow X_c$ from the sheared to canonical data.

The other decoder *DS*, takes as input the full latent code **z** and reconstructs the input sheared sample. In combination with the encoder $E$, it thus acts as an auto-encoder and learns a mapping $N : X_s \rightarrow X_s$ from the sheared data to itself.

Our objective function has three components:
(1) *L2 loss* for matching the generated canonical samples $M(x_t)$ to their targets $x_c$;
(2) *L1 patch loss* for matching random patches extracted from generated canonical images $M(x_t)$ and corresponding targets $x_c$ for enforcing image similarity locally as changes in sensor response due to change in contact conditions are predominately local in nature;
(3) *Reconstruction loss* for matching the generated sheared samples $N(x_t)$ to their target input sheared samples $x_s$.

The objective functions are:

$$\mathbb{L}_{\text{rec}}(E, DS) = \mathbb{E}_{x_s \sim X_s}[\|x_t - DS(z_p, z_s)\|_2], \tag{1}$$

$$\mathbb{L}_{\text{sup}}(E, DC) = \mathbb{E}_{x_s \sim X_s, x_c \sim X_c}[\|x_c - DC(z_p))\|_2], \tag{2}$$

$$\mathbb{L}_{\text{patch}}(E, DC) = \mathbb{E}_{x_s \sim X_s, x_c \sim X_c}[\|\text{crops}(x_c) - \text{crops}(DC(z_p)))\|_1], \tag{3}$$

where $(z_p, z_s) = E(x_s)$ for all $x_s \in X_s$. The function crops() selects a patch of size $20 \times 20$ pix from $M(x_t)$ and $x_c$. The objective functions $L_{\text{rec}}$, $L_{\text{sup}}$ and $L_{\text{patch}}$ represent sheared reconstruction loss, canonical supervised loss and canonical supervised patch loss respectively.

The final objectives for Encoder $E$ and Decoders $DC$, $DS$ are:

$$\mathbb{L}_E = \mathbb{L}_{\text{rec}}(E, DS) + \mathbb{L}_{\text{sup}}(E, DC) + \lambda \cdot \mathbb{L}_{\text{patch}}(E, DC), \tag{4}$$

$$\mathbb{L}_{DC} = \mathbb{L}_{\text{sup}}(E, DC) + \lambda \cdot \mathbb{L}_{\text{patch}}(E, DC), \tag{5}$$

$$\mathbb{L}_{DS} = \mathbb{L}_{\text{rec}}(E, DS), \tag{6}$$

where $\lambda = 0.1$ is the relative loss scale factor.

Details of the specific model architectures and their training are given in Appendices A & B.

## 4 Experimental Results

### 4.1 Shear Removal from Tactile Images

We first performed an ablation study to demonstrate the separation of latent representations by the *Disentangled UnshearNet* into pose and shear codes respectively. For details see Appendix D.

The first test of the effectiveness of our approach in removal of motion-induced global shear used multi-scale structural similarity index (SSIM) [26] to compare image similarity between *Disentangled (Entangled)* model-generated unsheared images *PC* and their test set sheared *S* and canonical (non-sheared) *C* counterparts. On comparison, we found the *Disentangled / Entangled UnshearNet* model generated unsheared images *PC* to be structurally closer to the target canonical images *C* (93% / 90%) than their sheared *S* counterparts (32%). This clearly shows that our model *Disentagled UnshearNet* performed better in removing global shear from tactile images *vis-á-vis* the baseline model *Entangled UnshearNet*.

### 4.2 Reconstruction of Local Contact Geometry

Motion-induced global shear masks the contact geometry necessitating removal of shear from tactile images to reconstruct true indentation field (Fig. 4, top row). We used a Voronoi-based method introduced in Cramphorn *et al* [13] to reconstruct indentation field from tactile data encoded by TacTip. It transforms the displacements of the tactile markers on the inner side of sensing surface into areas of hexagonal cells tessellating the grid of the markers [13, Fig. 4]. The change in Voronoi cell areas from an undeformed reference represent the magnitude of local skin distortion caused by contact, which correlates with the indentation due to contact geometry.

For display, we fit a 3D surface to the $(x, y)$ centroid coordinates using the Voronoi cell areas as the corresponding height values (Fig. 4, top row). The fitted surface to test set sheared data *S* is visibly distorted by shear (Fig. 4, top row). However, once the global shear is removed successfully by

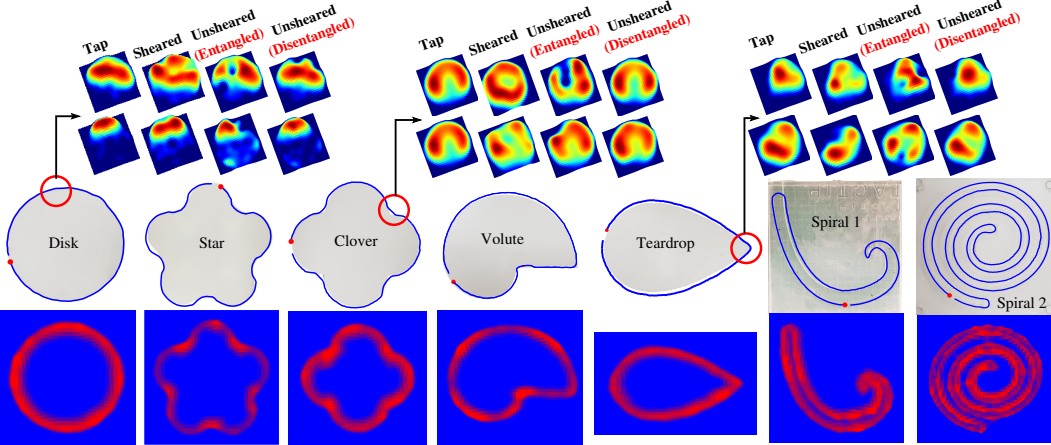

Figure 4: **Contact shape reconstructions, shape exploration and reconstruction.** Top row: Local contact geometry reconstruction. The approximate contact geometry can be recovered from sheared data using the model-generated unsheared images generated by *Disentangled UnshearNet* but not from the original sheared data. Middle row: Trajectories, overlaid on objects, under robust sliding using pose estimation from *Disentangled UnshearNet* generated unsheared images. Red dot is the starting point of trajectory. Bottom row: Full object reconstructions from combining the unsheared local contact geometry reconstructions along the sliding trajectories. High-resolution figures are in Appendix E.

*Disentagnled UnshearNet* in model-generated unsheared images *PC*, the fitted surface closely matches with the true representation of the indentation field obtained from the canonical (non-sheared / tap) data *C* (Fig. 4, top row). Also clear from Fig. 4 (top row) that our model *Disentangled UnshearNet* is more effective in removing global shear *vis-á-vis* the baseline model *Entangled UnshearNet*.

### 4.3  2D Sliding Shape Exploration

We further confirmed the performance of our model on removing global shear by computing the mean-absolute error (MAE) between the predicted pose (lateral position $\tau_X$ and in-plane orientation $\theta n_Z$) from *Disentangled / Entangled UnshearNet* generated unsheared images *PC*, sheared images *S* and canonical (non-sheared / tap) images *C* and the target pose using *PoseNet* trained only on canonical (non-sheared / vertical tap) images (Table 1). On comparison, the lowest prediction error was found on unsheared samples obtained using disentangled network. The error increased by as much as 50% and 5× higher on unsheared samples from entangled network and original sheared samples respectively. Again, the *UnshearNet* models managed to successfully correct the distortion in sensor response cause by slide-induced global shear, with the disentangled representation giving the best performance.

Table 1: Mean Absolute Error (MAE) in Pose Predictions

|  | Tap Data | Sheared Data | Unsheared Data (Entangled) | Unsheared Data (Disentangled) |
|---|---|---|---|---|
| horizontal, $\tau_x$ (mm) | 0.43 | 2.72 | 0.95 | 0.64 |
| yaw, $\theta n_Z$ (degrees) | 2.13 | 22.20 | 6.73 | 4.38 |

To test the real-time performance and model generalizability to novel stimuli, we used *PoseNet* for sliding exploration of several planar shapes (Fig. 4, second row). This testing included four additional planar shapes absent from training including two acrylic shapes that differed in frictional properties from the other 3D printed stimuli (spirals 1 & 2, Fig. 4, second row). The robot successfully traced contours across all stimuli shapes (Fig. 4, second row) using a combination of *Disentangled UnshearNet* for global shear removal, *PoseNet* for prediction of object pose from model-generated unsheared images and a simple servo control policy (see Appendix C). This clearly indicates that the proposed *Disentangled UnshearNet* is well suited for correcting the sliding-induced shear-distorted sensor response for real-time servo control applications.

## 4.4 Object Shape Reconstruction

As a final test of our model's utility to remove shear, we reconstructed full object shapes by combined use of contact geometry estimation (Sec. 4.3) and sliding shape exploration (Sec. 4.4), both tasks adversely affected by motion-induced shear (Fig. 4, third row). To attain full object reconstructions, we fused together the contact information extracted from unsheared images corresponding to sliding contacts (recorded during shape exploration) and linearly interpolated them over a rectangular grid (Fig. 4, third row). These results clearly demonstrate the effectiveness of our approach in learning the *sheared-unsheared* mapping that can then later be reused for multiple downstream tasks. In addition, the faithful shape reconstructions along with the shape exploration across multiple novel planar objects show that the model successfully generalizes to novel situations.

## 5  Discussion

This work proposed a supervised approach to remove the distortion in sensor response caused by motion-induced global shear while preserving sensor deformations induced by the spatial geometry of the contacted stimuli. The proposed approach was based on an encoder-decoder structure and learned to remove shear from tactile images by disentangling the pose (contact only) and shear in the latent space (Fig. 3). The disentanglement of sensor components contributed due to contact only and global shear promoted learning of robust *sheared-unsheared* mapping as demonstrated by better performance of this approach *vis-á-vis* the baseline (Fig. 2).

This approach was validated by: (1) demonstrating a good match (93%) between model-generated unsheared images (from sliding contacts) and its canonical (non-sheared / vertical tap) counterparts using structural similarity index measure (Sec 4.1). (2) faithful reconstruction of contact geometry, masked in sheared images, from model-generated unsheared images (Fig. 4, top row). (3) predicting the local object pose from model-generated unsheared images with a prediction network trained only on canonical (non-sheared / vertical tap) data (Table 1). This was later used for successful sliding exploration of several planar objects (Fig. 4, second row). 4) combining contact geometry reconstruction and sliding exploration to reconstruct full object shapes for several planar objects (Fig. 4, third row) which remains an problem under active investigation in the field [24, 25].

Our approach improves upon related work [6, 7] that instead trained insensitivity to shear into the prediction network itself. While effective, this approach does not extend readily to predict other tactile dimensions such as parameters of the local contact geometry. In that case, new labelled data would need collecting to train a new model for each intended task. Our approach, of shear removal from tactile images, instead means that a single model based on the canonical (tapping) data can be re-used.

The current study has several limitations: 1) considered only planar objects. We, however, expect it to extend readily to to 3D objects similar to recent results that have successfully extended *PoseNet* to complex 3D surfaces (3 pose components) and 3D edges (5 pose components) for sliding exploration [6, 7]. 2) tested only on TacTip optical tactile sensor. The approach should hold well for other soft optical sensors although difference in frictional and elastic properties between different sensor types will necessitate retraining of models. 3) training was limited to translation shear even though servo control introduces rotational shear due to sensor rotation while sliding over objects. The fact that the methods worked well both for sliding exploration and contact geometry reconstruction even in absence of explicit compensation for rotational shear, shows that some unshearing carries over from translation to rotational shear at least for planar objects. Our expectation is that bringing rotational shear into the training will become important when extending the methods to 3D objects. and 4) requires paired canonical and sheared data, which complicates the data collection. An approach that is more economical with labelled data requirements would be a worthwhile extension of the current work.

Overall, we expect the methods developed here can can be applied more widely to various other exploration and manipulation tasks involving soft tactile sensors that are adversely affected by motion-induced shear.

## Acknowledgements

Anupam Gupta and Nathan Lepora were supported by a Research Leadership Award from the Leverhulme Trust on 'A biomimetic forebrain for robot touch' (RL-2016-39).

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
