# OpenReview forum: "Tactile Image-to-Image Disentanglement of Contact Geometry from Motion-Induced Shear"
_robot-learning.org/CoRL/2021/Conference — CoRL2021 Poster_

### Official Review · Reviewer_xr1d · 2021-07-21

**Originality:** Good
**Technical Quality:** Very Good
**Clarity Of Presentation:** Good
**Impact:** 3

**Recommendation:**

Weak Accept: I recommend accepting the paper, but will not argue for my recommendation if the majority of other reviewers have a different opinion.

**Summary:**

This work proposed to disentangle the contact geometry from shear, which can benefit sliding motion and manipulation for soft tactile sensors. The authors proposed the encoder-decoder network trained from paired supervised data. The system is evaluated to reconstruct unsheared canonical images, local shape reconstruction. It is further applied for contour following with sliding motion.


**Issues:**

Quote from the weaknesses:
- The comparison between Entangled and Disentangled UnshearNet seems unfair. The Disentangled UnshearNet clearly has more supervision, in the autoencoder fashion to reconstruct the raw image. A fair comparison could be also adding raw input reconstruction to the Entangled network. By current comparison, it is unclear whether the improvement is from the additional supervision or the disentangled network flow.
- Additionally, there seems to be no clear evidence that the embedding of the Disentangled network is disentangled into “Pose Code” and “Shear Code”. What if the “Pose Code” contains both “deformation” and “shear” information, and “Shear Code” contains minor/part of shear information? It seems that the network flow doesn’t have the supervision to force the embedding to be disentangled, since the “Pose Code” is used for both geometry and raw input reconstruction. And there is no validation of what the “Shear Code” learns. One experiment would be using the “Pose Code” and “Shear Code” to predict the motion that induced the shear. It can also be added as the supervision to the network to force “Shear Code” alone to learn the motion.
- For 3.2.2 Pose-labelled data, the data collection procedure seems very similar to the one for canonical data. Is there a reason for collecting a separate dataset?


**Reviewer Expertise:**

Good: General knowledge of the area

**Strengths And Weaknesses:**

Strengths:
- The disengagement of the contact geometry and shear is meaningful, and can benefit future applications with shear-related tactile and manipulation tasks.
- The author proposed a network and automatic data collection framework to tackle the problem, using paired supervised data.
- The disentangled results are evaluated for contour following.

Weaknesses:
- The comparison between Entangled and Disentangled UnshearNet seems unfair. The Disentangled UnshearNet clearly has more supervision, in the autoencoder fashion to reconstruct the raw image. A fair comparison could be also adding raw input reconstruction to the Entangled network. By current comparison, it is unclear whether the improvement is from the additional supervision or the disentangled network flow.
- Additionally, there seems to be no clear evidence that the embedding of the Disentangled network is disentangled into “Pose Code” and “Shear Code”. What if the “Pose Code” contains both “deformation” and “shear” information, and “Shear Code” contains minor/part of shear information? It seems that the network flow doesn’t have the supervision to force the embedding to be disentangled, since the “Pose Code” is used for both geometry and raw input reconstruction. And there is no validation of what the “Shear Code” learns. One experiment would be using the “Pose Code” and “Shear Code” to predict the motion that induced the shear. It can also be added as the supervision to the network to force “Shear Code” alone to learn the motion.
- For 3.2.2 Pose-labelled data, the data collection procedure seems very similar to the one for canonical data. Is there a reason for collecting a separate dataset?


**Summary Of Recommendation:**

Overall, the paper demonstrated an important problem of entangled shear for contact geometry. The framework with the supervised paired data solved the problem well. But the technical part of the network structure requires more experiments to support its claim of disentanglement of “Pose Code” and “Shear Code” for the Disentangled UnshearNet.

---

> ### Author Response · Authors · 2021-08-28
> **Response to Reviewer-xr1d (part 1)**
>
> Thank you for your insightful comments on the manuscript, and recognizing that the problem we are addressing for ‘disengagement of the contact geometry and shear is meaningful, and can benefit future applications with shear-related tactile and manipulation tasks.’ We respond to your comments below.
>
> Comment: The comparison between Entangled and Disentangled UnshearNet seems unfair. The Disentangled UnshearNet clearly has more supervision, in the autoencoder fashion to reconstruct the raw image. A fair comparison could be also adding raw input reconstruction to the Entangled network. By current comparison, it is unclear whether the improvement is from the additional supervision or the disentangled network flow.
>
> Response: The ‘extra supervision’ used in Disentangled UnshearNet in the form of input (sheared) reconstruction has been used to enforce separation of latent representations. As discussed in response to the next comment, this network architectural choice successfully disentangles latent representations as desired. A baseline approach used to compare the performance of proposed disentangled approach would be one that does not disentangle – like the Entangled UnshearNet approach we have used as baseline. In our view, it would not give a better baseline to add input reconstruction to this approach, but we have tried your suggestion out of curiosity and so we can give a full response to your comment: we retrained the Entangled UnshearNet with input reconstruction as suggested. Overall, we observed no significant improvement evaluated via SSIM metrics between generated unsheared images PC and ground truth unsheared images C -- 90.61 (with input reconstruction) vs 90.35 (original, without input reconstruction). This shows that the reported improvement is due to disentanglement and not due to extra supervision in Disentangled UnshearNet.
>
> Comment: Additionally, there seems to be no clear evidence that the embedding of the Disentangled network is disentangled into “Pose Code” and “Shear Code”. What if the “Pose Code” contains both “deformation” and “shear” information, and “Shear Code” contains minor/part of shear information? It seems that the network flow doesn’t have the supervision to force the embedding to be disentangled, since the “Pose Code” is used for both geometry and raw input reconstruction. And there is no validation of what the “Shear Code” learns. One experiment would be using the “Pose Code” and “Shear Code” to predict the motion that induced the shear. It can also be added as the supervision to the network to force “Shear Code” alone to learn the motion.
>
> Response: This is a good point that we have taken completely on board to include two new tests that the disentanglement is working as we claimed:
>
> 1) We did an ablation study to verify the separation of latent representations in the Disentangled UnshearNet into pose and shear codes respectively. To do this, we passed the ‘Shear Code’ to the unsheared reconstruction decoder (DC) instead of the ‘Pose Code’. As expected, this led to a severe degradation in performance with MSE error between the ground truth images C (tap) and unsheared images PC increasing to 0.22, an order of magnitude higher than original of 0.023. Similarly, the SSIM index dropped to only 2% on use of 'Shear Code' to reconstruct PC instead of the original 93% when 'Pose Code' was used.
>
> 2) In the similar fashion, we investigated the impact of replacing the ‘Shear Code’ with ‘Pose Code’ to reconstruct the sheared input S. If the network fails to disentangle the latent representation, with ‘Pose Code’ encoding the majority of information regarding both contact geometry and motion-induced shear, then the test metrics (SSIM and MSE error) should remain unchanged. However, the MSE error between S and the sheared output PS increased to 0.1, which is 50 times the original of 0.002 when both 'Pose and Shear Codes' were used to reconstruct PS. The SSIM index showed a similar trend, with the similarity dropping from 99.5% to only 11%. This shows that the ‘Shear Code’ is indeed encoding relevant information required for successful reconstruction of sheared input S.
>
> The above results demonstrate that the Disentangled UnshearNet successfully disentangles the latent representations as desired. The results of both of these new studies have been included in the manuscript  in Appendix D.

---

> > ### Author Response · Authors · 2021-08-28
> > **Response to Reviewer-xr1d (part 2)**
> >
> > Comment: For 3.2.2 Pose-labelled data, the data collection procedure seems very similar to the one for canonical data. Is there a reason for collecting a separate dataset?
> >
> > Response: The Pose-labelled data was collected to train PoseNet instead of reusing the previously collected pose data to train Entangled and Disentangled UnshearNet. This was done to ensure that the trained UnshearNet models work despite potential dataset shifts -- due to a host of factors including drift in sensor characteristics over time or with continued use due to wear and tear and any other potentially uncontrolled experimental factors -- and not overfit to training dataset. This has been clarified in the text in the lines 161-164, section 3.2.2.

---

> > > ### Comment · Reviewer_xr1d · 2021-09-03
> > > **Reply to author**
> > >
> > > Thank you for the response!
> > > Appreciate the authors for the ablation studies. These experiments make it more convincing for the disentanglement of "pose" and "shear", and show the effectiveness of the proposed network. I've updated my rating.

---

### Official Review · Reviewer_oHk8 · 2021-07-21

**Originality:** Good
**Technical Quality:** Good
**Clarity Of Presentation:** Good
**Impact:** 3

**Recommendation:**

Weak Accept: I recommend accepting the paper, but will not argue for my recommendation if the majority of other reviewers have a different opinion.

**Summary:**

The paper presents a method for handling the motion-dependent shear effect in tactile sensors. In particular, as objects can move under soft tactile sensors, tactile signal is distorted or sheared, which leads to difficulties in employing the resulting tactile signal, for example when performing object pose estimation. The paper proposes to learn “unshearing” deep neural network models that can map from sheared tactile images to canonical non-sheared images. Two architectures are proposed: entangled architecture that directly learns to reconstruct the canonical image from the sheared one, and a disentangled architecture that decomposes the intermediate representation into canonical and shear-specific parts that are trained to reconstruct the canonical and the original sheared tactile images respectively. Finally, the unsheared images are used to estimate object poses using a pose-model trained on the canonical non-sheared images. The paper collects a paired dataset of sheared and canonical tactile data to train the unshearing models. The resulting models are evaluated on three objects and TacTip biomimetic tactile sensor. They are shown to produce unsheared tactile images with a high similarity to the canonical ones and successfully reconstruct the original geometry of the objects, as well as significantly improve object pose prediction, with the disentangled model having a higher performance than the entangled one. Pose prediction is also used to perform servo control of robot following contours of the objects.

**Issues:**

Please refer to "Strengths And Weaknesses" for a detailed list of issues.

**Reviewer Expertise:**

Good: General knowledge of the area

**Strengths And Weaknesses:**

Strengths:
- Reducing the shearing effect of tactile sensors is important for real-world application of tactile signals.
- Disentangling reconstruction into sheared and non-sheared parts is an elegant way to improve unshearing performance.
- The paper is well-written and easy to understand and follow.
- The method is shown to work on a real TacTip biomimetic tactile sensor and produce a high level of reconstruction accuracy.

Weaknesses:
- Although the dataset is split into training and test parts, all the data still comes from the same three objects. It would be useful to see performance of the method on some novel objects to better understand the generalization capability of the method.
- Right now, the PoseNet is trained on canonical images and evaluated on sheared/unsheared images. It would be interesting to see results of training the PoseNet directly on the sheared images if such a pairing is possible. This would help to better understand the advantage of reconstructing the unsheared image for pose prediction.
- It would be interesting to see more details on the robotic control system used to perform the contour following experiments.

**Summary Of Recommendation:**

The paper shows that it is possible to unshear tactile sensor signals, which is important for many real-world robotic applications. The paper could benefit from some additional experiments to demonstrate the generalization capabilities of the method.

---

> ### Author Response · Authors · 2021-08-28
> **Response to Reviewer-oHk8**
>
> Thank you for your insightful comments. We are pleased to see your recognition that ‘this problem is important for real-world application of tactile signals’ and that ‘disentangling reconstruction into sheared and non-sheared parts is an elegant way’ to address this problem. We have revised the manuscript in response to the weaknesses you have identified, specifically around ‘additional experiments to demonstrate the generalization capabilities of the method.’
>
> Comment: Although the dataset is split into training and test parts, all the data still comes from the same three objects. It would be useful to see performance of the method on some novel objects to better understand the generalization capability of the method.
>
> Response: A similar point has been raised by multiple reviewers and the Area Chair, and we have fully taken this on board in the revised manuscript. In particular, to demonstrate further generalizability of the model to novel shapes, we have updated figure 4 in the manuscript with 4 complex shapes not encountered in training (including two acrylic shapes with distinct frictional properties to the 3D-printed shapes).
>
> Please also note the training data from the original three shapes was collected at a single location (Figure 1, red circle; see caption and text line 133). Therefore, some generalization had already been demonstrated as 2 shapes (the clover and teardrop) were non-uniform. This has been clarified further in the text in lines 117-119. section 3.1.
>
> Comment: Right now, the PoseNet is trained on canonical images and evaluated on sheared/unsheared images. It would be interesting to see results of training the PoseNet directly on the sheared images if such a pairing is possible. This would help to better understand the advantage of reconstructing the unsheared image for pose prediction.
>
> Response: PoseNet can indeed be trained to predict object pose directly from sheared images, an approach pursued by Refs [5, 6, 7]. A qualitative comparison reveals that both approaches, ours and task-specify approach pursued by Refs [5, 6, 7], enable successful smooth servo control over a wide array of planar objects. Our approach, however, provide several advantages over the task-specific approach. These have been discussed before as response to comments by Area Chair; our full response is there, but we summarize our view of the main advantages of our approach of: (a) being computationally and data efficient requiring the ‘unshearing’ of tactile data to be learned once instead of relearning for each downstream task like object pose and contact reconstruction for full object reconstruction; (b) the models can then be learnt more easily on unsheared data (as we demonstrated on pose) or in some cases no model need to be learnt at all (as in our contact surface reconstruction using a Voronoi tessellation); and (c) is more easily extensible to including other downstream variables of interest like indentation depth.
>
> Comment: It would be interesting to see more details on the robotic control system used to perform the contour following experiments.
>
> Response: We have now included the details on the control system used for servo experiments in Appendix C.

---

> > ### Comment · Reviewer_oHk8 · 2021-09-03
> > **Thanks**
> >
> > Thanks for addressing my comments and including additional experiments. I have updated my rating.

---

### Official Review · Reviewer_JxBB · 2021-07-24

**Originality:** Fair
**Technical Quality:** Good
**Clarity Of Presentation:** Very Good
**Impact:** 2

**Recommendation:**

Weak Reject: I recommend rejecting the paper, but will not argue for my recommendation if the majority of other reviewers have a different opinion.

**Summary:**

The paper proposes a supervised learning method to remove artifacts from optical-tactile signals for robot manipulation tasks. Specifically, the artifacts in question are caused by sensor distortion in the presence of motion-induced shear. In the approach, the model is trained on tactile images collected from un-sheared contact with a few distinct canonical shapes.

**Issues:**

Please address the points listed above.

**Reviewer Expertise:**

Very good: Comprehensive knowledge of the area

**Strengths And Weaknesses:**

Strengths:
- The paper is generally well written and easy to follow
- Experiments are detailed and the approach is clearly defined.
- Results seem promising.

Weaknesses
- The scope of the paper strikes me as fairly narrow, both in experiment and comparison to existing methods.
- Why does the approach not consider task-specific training, such as predicting local surface geometry directly for surface mapping (as opposed to an intermediate un-sheared representation) ? It would be nice to highlight the benefits of this approach when compared to approaches which model local contact geometry, such as the following (I have no affiliation):
- Bauza, Maria, et al. "Tactile object pose estimation from the first touch with geometric contact rendering." arXiv preprint arXiv:2012.05205 (2020).
- There seems to be a lack of baselines in the approach. How might this compare to using an analytical model, for instance?
- The scope of the experiments seem quite narrow. Only three shapes were investigated, and all were made from the same material. To demonstrate that the method is indeed learning a meaningful representation (as opposed to over-fitting to data), it would be useful to use test objects and motions which were not observed in training.
- There does not seem to be any discussion of rotational vs translational shear. I would think that the former would be more challenging to disentangle than the latter, but this should be discussed and investigated.
-  A description and picture of the TacTip sensor would be very helpful to the reader.


**Summary Of Recommendation:**

A significant contribution to the community would require broadening the scope of the paper, clearly demonstrating the generality of the approach comparison to existing state-of-the-art methods.

---

> ### Author Response · Authors · 2021-08-28
> **Response to Reviewer-JxBB (part 1)**
>
> Thank you for taking the time to provide thorough comments on our manuscript. We can see you recognize some positives in terms of the clarity and promising results but have concerns e.g around the scope. In response, we have significantly extended the paper to address these concerns.
>
> Comment: The scope of the paper strikes me as fairly narrow, both in experiment and comparison to existing methods
>
> Response: A similar point has been raised by multiple reviewers and the Area Chair. In response, the paper has now been extended in multiple ways:
>
> (1) To demonstrate further generalization of the model to novel shapes, we have updated figure 4 in the manuscript with 4 complex shapes not encountered in training (including two acrylic shapes with distinct frictional properties to the 3D-printed shapes).
> That said, the training data from the original three shapes was collected at a single location (Figure 1, red circle; see caption and text line 133). Therefore, some generalization had already been demonstrated as 2 shapes (the clover and teardrop) were non-uniform. This has been clarified further in the text in lines 117-119.
>
> (2) We have now extended results to include full object reconstruction for all 7 shapes in figure 4, in addition to local contact reconstruction results for training shapes (test data) and full servo results also for all 7 shapes. The contour following results along with full object reconstruction jointly demonstrate the generalizability and effectiveness of our approach to remove shear from novel contacts. In addition, they also demonstrate the motivation behind our approach to unshear the raw sensor response, later used for multiple downstream tasks obviating the need to learn unshearing of tactile data repeatedly for each downstream task, making it computationally and data efficient. This has been further clarified in the text in section 4.4.
>
> (3) In addition, a qualitative comparison between our approach of pose prediction for smooth servoing over planar objects to the ones pursued by Refs [5, 6, 7] that learn to directly predict object pose from sheared images (by learning to ignore motion-induced shear) shows that both approaches allow successful smooth servoing across a wide array of object shapes. However, our approach has several advantages in: (a) being computationally and data efficient by only requiring the ‘unshearing’ of the tactile data to be learnt once; (b) the models for downstream tasks can then be learnt more easily on unsheared data (as we demonstrated on pose) or in some cases no model need to be learnt at all (as in our contact surface reconstruction using a Voronoi tessellation); and (c) is more easily extensible to including other downstream variables of interest like indentation depth. This has been mentioned in the text in the lines 34-43 in section 1.
>
> In terms of a baseline for contact surface reconstruction and full object reconstruction, it seems more direct to compare with the actual object shape itself, which we have done in figure 4. There are various other approaches in the literature for object reconstruction dating back to original work in the 1980s (e.g. Allen, IJRR 1988); however, these approaches tend to be tactile sensor specific, which does not facilitate a direct comparison, and is somewhat tangential to the focus of the paper on removing shear.

---

> > ### Author Response · Authors · 2021-08-28
> > **Response to Reviewer-JxBB (part 2)**
> >
> > Comment: Why does the approach not consider task-specific training, such as predicting local surface geometry directly for surface mapping (as opposed to an intermediate un-sheared representation)?
> >
> > This comment has some overlap with the point raised by the Area Chair on our overall perspective; our full response is given there, but we summarize our view of the main advantages of a disentanglement approach of: (a) being computationally and data efficient by only requiring the ‘unshearing’ of the tactile data to be learnt once; (b) the models can then be learnt more easily on unsheared data (as we demonstrated on pose) or in some cases no model need to be learnt at all (as in our contact surface reconstruction using a Voronoi tessellation); and (c) is more easily extensible to including other downstream variables of interest.
> >
> > Also, our method for surface geometry reconstruction directly uses a Voronoi tessellation of the tactile data, which has benefits in not having to train a model. As shown in Figure 4, this method would not work unless the tactile data has been preprocessed to remove shear.
> >
> > Comment: It would be nice to highlight the benefits of this approach when compared to approaches which model local contact geometry, such as the following (I have no affiliation):
> > Bauza, Maria, et al. "Tactile object pose estimation from the first touch with geometric contact rendering." arXiv preprint arXiv:2012.05205 (2020).
> >
> > Response: This is a good paper but its aims and methods are distinct from the present study: its aims are to infer the global pose of a known object using an object model and tactile contacts. Our paper is tackling a distinct problem of estimating local geometric features of an object free of any shear affects due to the contact. There is some overlap in the new results in the revised manuscript on reconstructing the object shape from the contact (Figure 4), which also gives pose information about the object; however, we are making no prior assumptions about object shape, and as reviewed in Luo et al, Mechatronics 2017, there has been a fair amount of research on object pose estimation whereas to the best of our knowledge there has been no previous work on disentanglement approaches for shear removal in tactile sensing.
> > The paper makes a nice reference to related work in the field, and so we have cited it in the discussion.
> >
> > Comment: The scope of the experiments seem quite narrow. Only three shapes were investigated, and all were made from the same material. To demonstrate that the method is indeed learning a meaningful representation (as opposed to over-fitting to data), it would be useful to use test objects and motions which were not observed in training.
> >
> > Response: We agree completely and have significantly extended our experimental work to include more variety of shapes and different materials, as described in response (1) above to your first comment.
> >
> > Comment: There does not seem to be any discussion of rotational vs translational shear. I would think that the former would be more challenging to disentangle than the latter, but this should be discussed and investigated.
> >
> > Response: For simplicity, we considered only translational shear in training. However, the servo control experiments for object shape reconstruction necessarily introduce rotational shear because the sensor rotates while in contact with the object. That the methods still work well for servo control and reconstruction, as shown by the results in figure 4, is evidence that some unshearing carries over from translational to rotational shear, which is a finding we have now emphasized in the lines 298-304, section 5.
> >
> > We expect the current methods would extend straightforwardly to training also with rotational shear, given these findings. However, we would not expect much improvement on the current task using planar objects as the performance is already very good. In our view, including rotational shear would be most appropriately covered in extensions of this work to 3d objects, which is a natural direction to take this work in the future.
> >
> > Comment: A description and picture of the TacTip sensor would be very helpful to the reader.
> >
> > Response: We have added a picture of TacTip along with an internal view of the sensing surface with exposed internal markers in figure 1 (a) & (b). Also, the details on the sensor operation are provided in the lines 103-114, section 3.1.

---

> > > ### Comment · Reviewer_JxBB · 2021-09-03
> > > **Reply to Authors' response.**
> > >
> > > I appreciate the efforts made to address each of my questions and including modifications to the manuscript. I think the quality of the paper has improved as a result, and will be increasing my score.

---

### Official Review · Reviewer_gcAq · 2021-07-25

**Originality:** Good
**Technical Quality:** Good
**Clarity Of Presentation:** Good
**Impact:** 4

**Recommendation:**

Weak Accept: I recommend accepting the paper, but will not argue for my recommendation if the majority of other reviewers have a different opinion.

**Summary:**

The paper develops a learning paradigm to obtain non sheared tactile images from shear contact.

**Issues:**

Addressing the below issues will improve the paper:

1. It’s not clear whether the model was trained separately for each object or if a single model was trained across objects.

2. Running experiments on unseen objects.


**Reviewer Expertise:**

Very good: Comprehensive knowledge of the area

**Strengths And Weaknesses:**

The paper is attempting to solve an interesting problem which benefits the domain of visuotactile sensors for tactile perception. The illustrations in figures bring clarity to the approach. There are some concerns with the paper in its current form which are listed below:

1. One major concern with the approach is that it’s not clear to me whether the approach works on unseen objects. Also, it was not clear whether the learned model is object dependent. Conducting experiments on unseen objects would help bring clarity.

2. Does the model generalize across sensors or do we need to train per sensor?

3. Additionally, the paper mentions an appendix but it was not submitted.



**Summary Of Recommendation:**

I would like to know whether the model generalizes before I can make a better recommendation.

---

> ### Author Response · Authors · 2021-08-28
> **Response to Reviewer-gcAq**
>
> Thank you for your insightful comments on the manuscript and for your appreciation that our paper is ‘attempting to solve an interesting problem which benefits the domain of visuotactile sensors’. We recognize your concerns and have revised the paper to address them.
>
> Comment: One major concern with the approach is that it’s not clear to me whether the approach works on unseen objects. Also, it was not clear whether the learned model is object dependent. Conducting experiments on unseen objects would help bring clarity.
>
> Response: We agree completely, and a similar point has been raised by multiple reviewers and the Area Chair. We repeat below the response we gave to the Area Chair:
>
> To address the concern regarding the generalization of the model to novel shapes, we have updated figure 4 in the manuscript with 4 complex shapes not encountered in training (including two acrylic shapes with distinct frictional properties to the 3D-printed shapes).
> Please also note that the training data from the original three shapes was collected at a single location (Figure 1 (c), red circle; see caption and text lines 133). Therefore, some generalization had already been demonstrated as 2 shapes (the clover and teardrop) were non-uniform. This has been clarified further in the text in lines 117-119.
>
>
> Comment: Does the model generalize across sensors or do we need to train per sensor?
>
> Response: This question can be interpreted in two ways (a) generalize across sensors of the same design e.g. two TacTips with small differences due to hand fabrication; and (b) generalize across distinct sensor designs e.g. a TacTip and GelSight.
>
> On (a), all data, for all shapes including the 4 new complex shapes, for 2d servoing and full object reconstructions (second and third row, figure 4) were collected using a newly-fabricated sensor rather than the one used for training. The final assembly of sensors is done by hand that introduces variability in the response in other otherwise same sensors, which has been mentioned in the text in lines 111 - 113, section 3.1.
>
> On (b), we do not have access to other tactile sensors so cannot test this directly. The methods should apply to other optical tactile sensors but we would expect a new model would need training, as the unshearing relates to the frictional and elastic properties of the sensor which will differ across sensor types. This has been clarified in the lines 294-297, section 5.
>
> Comment: Additionally, the paper mentions an appendix but it was not submitted.
>
> Response: The appendix is attached directly to the end of the paper. We think the confusion likely stemmed from the fact that the appendix was not submitted as a separate document at the time of manuscript submission.
>
> Comment: It’s not clear whether the model was trained separately for each object or if a single model was trained across objects.
>
> Response: The same single model was trained across all objects. This has been clarified in the text in the lines 405-406, Appendix B.

---

> > ### Comment · Reviewer_gcAq · 2021-09-04
> > **Thanks (updated rating)**
> >
> > Thanks for the clarifications. I have updated the scores to reflect the rebuttal.

---

### Meta-Review · Area_Chair_vsFX · 2021-08-16

**Recommendation:** Accept (Poster)
**Confidence:** 4

**Metareview:**

This paper presents a machine learning model to disentangle the effects of motion-induced shear from contact geometry in vision-based tactile sensors.

Quality:

(-) All reviewers have concern that the experimental validation does not fully support the claims of the manuscript. I here highlight the concerns from Reviewer gcAq and JxBB regarding the generalization capabilities of the proposed approach to unseen objects and sensors, and the additional experiments suggested by Reviewer xr1d to validate the claims about the Disentangled UnshearNet. Performing the experiments requested by the four reviewers would benefit the overall quality of the paper.

(+) Real-world experiments

Clarity:

(+) The paper is generally clear and well written

(/) The appendix is not included

Originality:

(+) The proposed approach is novel

Significance:

(+) Several reviews agree that the topic is relevant and impactful

Additional comment: From a philosophical perspective, it should be better argued why Disentangling Contact Geometry from Motion-Induced Shear is useful. If you consider that this is a component of a wider system (e.g., control), a devil's advocate might ask why not using an end-to-end learning approach (together with the downstream task) in the first place and side-step the need to explicitly disentangle.


---

After rebuttal:
The authors included additional experiments and answered the reviewers in such a way as to address most of the concerns.
While the paper is likely to impact a very niche part of the robotic community, the contribution appears to be solid.

---

> ### Author Response · Authors · 2021-08-28
> **Response to AC-vsFX (part 1)**
>
> Thank you for your summary of the main points of the reviewer comments, and your insights into raising the relative merits of end-to-end versus a disentanglement approach.
>
> Comment:  All reviewers have concern that the experimental validation does not fully support the claims of the manuscript. I here highlight the concerns from Reviewer gcAq and JxBB regarding the generalization capabilities of the proposed approach to unseen objects and sensors, and the additional experiments suggested by Reviewer xr1d to validate the claims about the Disentangled UnshearNet. Performing the experiments requested by the four reviewers would benefit the overall quality of the paper.
>
> Response: We agree completely that the experimental validation needed improvement. Fortunately, some of these analyses had been carried out while the paper was in review, so we have been able to extend the experimental sections of the paper to address all reviewer comments. Specifically:
>
> 1) To address the concern regarding the generalization of the model to novel shapes, we have updated figure 4 in the manuscript with 4 complex shapes not encountered in training (including two acrylic shapes with distinct frictional properties to the 3D-printed shapes).
> Please also note that the training data from the original three shapes was collected at a single location (Figure 1 (c), red circle; see caption and text line 133). Therefore, some generalization had already been demonstrated as 2 shapes (the clover and teardrop) were non-uniform. This has been clarified further in the text in lines 117-119, section 3.1.
>
> 2) One reviewer asked if the methods generalize from one sensor to another. All data, for all shapes including the 4 new complex shapes, for 2d servoing and full object reconstructions (second and third row, figure 4) was collected using a newly-fabricated sensor rather than the one used for training. The final assembly of sensors is done by hand that introduces variability in the response in other otherwise same sensors, which has been mentioned in the text in lines 111 - 113, section 3.1.
>
> 3) We have also added full object reconstructions for all seven object shapes to demonstrate the generalization of our model to novel contacts (third row, figure 4).
>
> 4) In addition, we have included a new ablation study (Appendix D) to demonstrate the successful disentanglement of latent representation into ‘Pose Code’ and ‘Shear Code’ by Disentangled UnshearNet which was questioned by reviewer xr1d.
>
> Comment: (/) The appendix is not included
>
> Response: The appendix is attached directly to the end of the paper. We think the confusion likely stemmed from the fact that the appendix was not submitted as a separate document at the time of manuscript submission.

---

> > ### Author Response · Authors · 2021-08-28
> > **Response to AC-vsFX (part 2)**
> >
> > Comment: From a philosophical perspective, it should be better argued why Disentangling Contact Geometry from Motion-Induced Shear is useful. If you consider that this is a component of a wider system (e.g., control), a devil's advocate might ask why not using an end-to-end learning approach (together with the downstream task) in the first place and side-step the need to explicitly disentangle.
> >
> > Response: This is an interesting comment, which we agree we should articulate a good answer. We see the following reasons for a disentangled versus end-to-end approach:
> >
> > 1) Disentangling environmental factors of variation and object attributes has received no attention in touch but has been argued in computer vision as important to improve the generalization of machine learning models to unobserved situations. The disentangling/separation of irrelevant environmental factors from object attributes can aid in learning of robust object representations, independent of environmental factors, important for object classification under novel scenarios. Furthermore, once disentangled, different environmental factors and object attributes can be recombined to generate coherent novel concepts thus extending knowledge to previously unobserved scenarios. For example, synthesizing sensor responses to novel stimuli by combining the stimuli attributes and factor of variation without actually observing them to plan action under previously unobserved conditions. In the case of tactile sensing considered here, the stimuli attributes could be novel contact geometry and the factors of variation the motion-induced shear. This has been mentioned in the manuscript in the lines 90 - 99, section 2.
> >
> > 2) In tasks involving robot touch to explore objects to reconstruct their shape, it is necessary to predict the sensor pose (relative to the local object) to control the sensor while also inferring the contact geometry for the reconstruction. However, for soft tactile sensors, both outputs are adversely affected by motion-induced shear. The options are to: A) remove the motion-induced shear component from sensor response first, then use the unsheared sensor response to predict pose and other geometric variables for downstream tasks or B) train separate models for each of the downstream tasks in an end-to-end fashion that predict output variables directly from sheared sensor response or a more complex multi-headed model where each head predicts output variables corresponding to an individual downstream task.
> > The approach taken in our paper (option A) has advantages in: (a) being computationally and data efficient by only requiring the ‘unshearing’ of the tactile data to be learnt once instead of relearning for each tactile task, such as when applied to full object reconstruction; (b) the models can then be learnt more easily on unsheared data (as we demonstrated on pose) or in some cases no model need be learnt at all (as in our contact surface reconstruction using a Voronoi tessellation); and (c) is more easily extensible to including other downstream variables of interest, for example, indentation depth. This has been mentioned in the text in the lines 34-43 in section 1.

---

### Decision · Program_Chairs · 2021-09-13

**Decision:**

Accept (Poster)

**Comment:**

This paper presents a machine learning model to disentangle the effects of motion-induced shear from contact geometry in vision-based tactile sensors.

Quality:

(-) All reviewers have concern that the experimental validation does not fully support the claims of the manuscript. I here highlight the concerns from Reviewer gcAq and JxBB regarding the generalization capabilities of the proposed approach to unseen objects and sensors, and the additional experiments suggested by Reviewer xr1d to validate the claims about the Disentangled UnshearNet. Performing the experiments requested by the four reviewers would benefit the overall quality of the paper.

(+) Real-world experiments

Clarity:

(+) The paper is generally clear and well written

(/) The appendix is not included

Originality:

(+) The proposed approach is novel

Significance:

(+) Several reviews agree that the topic is relevant and impactful

Additional comment: From a philosophical perspective, it should be better argued why Disentangling Contact Geometry from Motion-Induced Shear is useful. If you consider that this is a component of a wider system (e.g., control), a devil's advocate might ask why not using an end-to-end learning approach (together with the downstream task) in the first place and side-step the need to explicitly disentangle.


---

After rebuttal:
The authors included additional experiments and answered the reviewers in such a way as to address most of the concerns.
While the paper is likely to impact a very niche part of the robotic community, the contribution appears to be solid.